**DOI: 10.1038/ncomms15929**　　**OPEN**

# Jahn-Teller distortion driven magnetic polarons in magnetite

H.Y. Huang[1,2], Z.Y. Chen[3], R.-P. Wang[4], F.M.F. de Groot[4], W.B. Wu[1], J. Okamoto[1], A. Chainani[1], A. Singh[1], Z.-Y. Li[5], J.-S. Zhou[5], H.-T. Jeng[3], G.Y. Guo[6,7], Je-Geun Park[8,9], L.H. Tjeng[10], C.T. Chen[1] & D.J. Huang[1,3]

The first known magnetic mineral, magnetite, has unusual properties, which have fascinated mankind for centuries; it undergoes the Verwey transition around 120 K with an abrupt change in structure and electrical conductivity. The mechanism of the Verwey transition, however, remains contentious. Here we use resonant inelastic X-ray scattering over a wide temperature range across the Verwey transition to identify and separate out the magnetic excitations derived from nominal $Fe^{2+}$ and $Fe^{3+}$ states. Comparison of the experimental results with crystal-field multiplet calculations shows that the spin–orbital $dd$ excitons of the $Fe^{2+}$ sites arise from a tetragonal Jahn-Teller active polaronic distortion of the $Fe^{2+}O_6$ octahedra. These low-energy excitations, which get weakened for temperatures above 350 K but persist at least up to 550 K, are distinct from optical excitations and are best explained as magnetic polarons.

[1] National Synchrotron Radiation Research Center, Hsinchu 30076, Taiwan. [2] Program of Science and Technology of Synchrotron Light Source, National Tsing Hua University, Hsinchu 30013, Taiwan. [3] Department of Physics, National Tsing Hua University, Hsinchu 30013, Taiwan. [4] Inorganic Chemistry and Catalysis, Utrecht University, Universiteitsweg 99, 3584 CG Utrecht, The Netherlands. [5] Department of Mechanical Engineering, Texas Material Institute, University of Texas at Austin, Austin, Texas 78712, USA. [6] Department of Physics, National Taiwan University, Taipei 10617, Taiwan. [7] Division of Physics, National Center for Theoretical Sciences, Hsinchu 30013, Taiwan. [8] Department of Physics and Astronomy, Seoul National University, Seoul 08826, Korea. [9] Center for Correlated Electron Systems, Institute for Basic Science, Seoul 08826, Korea. [10] Max Planck Institute for Chemical Physics of Solids, Nöthnitzerstr. 40, 01187 Dresden, Germany. Correspondence and requests for materials should be addressed to D.J.H. (email: djhuang@nsrrc.org.tw).

Since its first X-ray structural elucidation by Bragg[1] a hundred years ago and the discovery of the Verwey transition[2,3], magnetite ($Fe_3O_4$), has received much attention for decades. Even today, it attracts significant scientific and technological interest for its applications in ultrafast magnetic sensors[4], palaeomagnetism[5], nanomedicine carriers[6], and so on. $Fe_3O_4$ becomes ferrimagnetic below $T_c \sim 850\,K$, followed by an abrupt decrease in its electrical conductivity by two orders of magnitude as the temperature is cooled below $T_V$. In this first known magnet to mankind, one-third of Fe sites, termed $A$-sites, are tetrahedrally ($T_d$) coordinated with oxygens; the other two-thirds, termed $B$-sites, have octahedral ($O_h$) coordination. Verwey first suggested a $Fe^{2+}$–$Fe^{3+}$ charge-ordering occurring on the $B$-sites as the driving force of this transition. There are two major schools of interpretation: the first one interprets the Verwey transition as a transition driven by charge/orbital ordering[7–17]. The second one exploits the mechanism of a lattice distortion-driven electron–phonon coupling[18–22] enhanced by the on-site Coulomb interaction and thus opens a gap at the Fermi energy when the temperature is lowered below the Verwey transition temperature $T_V$.

Although numerous investigations have been carried out to verify the charge localization on the $B$-sites, the charge-ordering pattern of magnetite is subtle and still elusive[19,20]. While it is agreed that the charge disproportionation involves changes in the nominal $Fe^{2+}$ and $Fe^{3+}$ states associated with the $B$-sites, X-ray diffraction studies of the low-temperature phase of magnetite microcrystals[15,17] revealed that the $t_{2g}$ electrons of the $B$-sites are not fully localized in the form of $Fe^{2+}$ states. Instead, the electrons are distributed over linear three-Fe-site units termed trimerons, which are coupled to the $T_d$ Jahn-Teller distortion of $B$-site $Fe^{2+}O_6$ octahedra, as illustrated in Fig. 1. To the first approximation, the $B$-site $Fe^{3+}O_6$ octahedra are Jahn-Teller-inactive. The tetragonal distortion of $B$-site $Fe^{2+}O_6$ octahedra removes the degeneracy of $t_{2g}$ orbitals, in going from $O_h$ symmetry to $D_{4h}$ symmetry. In the absence of spin–orbit coupling, an effective energy separation $\Delta_{t_{2g}}$ between $d_{xy}$ and $d_{yz}/d_{zx}$ is created if the four Fe–O bonds in the $xy$ plane are elongated or contracted. The trimeron scenario then indicates that the Verwey transition is essentially due to an ordering of trimerons. Because previous results of optical conductivity[23] and photoemission[24–27] showed the pseudogap feature of magnetite above $T_V$, and results of entropy analysis[28], neutron/X-ray diffuse scattering[29] and anomalous phonon broadening[22] revealed the short-range order above $T_V$, one important open question is whether trimeron correlations persist in the cubic phase at temperatures above $T_V$. Combining these short-range correlations of polaronic characters with the spin degrees of freedom of $t_{2g}$ electrons, one can expect magnetic polarons in magnetite.

Here we present measurements of resonant inelastic X-ray scattering (RIXS)[30,31] at the Fe $L_3$-edge on magnetite to reveal the low-energy spin–orbital excitations of $Fe^{2+}$ ions in both the monoclinic and cubic phases. To the best of our knowledge, the magnetic excitations derived from the local tetragonal distortion field of $Fe^{2+}$ ions, that is, magnetic polarons, have not been reported to date. In combination with crystal-field multiplet calculations, we show the the existence of magnetic polarons in magnetite which is driven by the Jahn-Teller distortion.

## Results

**Fe $L_3$-edge RIXS.** Figure 2a shows the Fe $L$-edge X-ray absorption spectrum of magnetite. By comparing with crystal-field multiplet calculations (see Supplementary Fig. 1), it is understood that the absorption-energy centroid of $Fe^{2+}$ ions is lower than that of

$Fe^{3+}$ ions by $\sim 1$–$2\,eV$, consistent with earlier work[32–34]. Accordingly, the features at X-ray energies of 706.0 and 707.5 eV originate from the absorption of octahedrally coordinated $B$-site $Fe^{2+}$ states, while the maximum intensity feature at 708.8 eV is dominated by absorption from the the $Fe^{3+}$ ions of both the $B$-site octahedral and $A$-site tetrahedral symmetries.

The colour map of RIXS intensity measured at 80 K in the plane of incident photon energy versus energy loss shown in Fig. 2b presents the evolution of the RIXS spectral profile associated with $Fe^{2+}$ and $Fe^{3+}$ ions as detailed in the following. When the incident X-ray energy was set to below 707.5 eV, we observed $dd$ excitations of $Fe^{2+}$ with energy losses at $2.8 \pm 0.05$, $1.65 \pm 0.05$ and $1.16 \pm 0.05\,eV$ shown in Fig. 2c, and also a broad excitation centred at 200 meV shown in Fig. 2d. If the incident X-ray energy goes beyond 707.5 eV, the 1.16-eV $dd$ excitation of $Fe^{2+}$ begins to evolve into a fluorescence that has a constant X-ray emission energy independent of incident energy. With the incident X-ray energy set to 708.8 eV, RIXS excitations arise mostly from $Fe^{3+}$ ions of octahedral or tetrahedral symmetry.

Figure 2d shows two RIXS features centred at 90 and 200 meV in a magnified plot of energy loss below 0.7 eV. Measurements carried out by varying the scattering angle suggested that these two low-energy excitations do not disperse in momentum space (see Supplementary Fig. 2). The 200-meV excitation has a full-width at half-maximum larger than the instrumental energy resolution. This broad RIXS feature resonates near the $L_3$-edge of $Fe^{2+}$ and almost disappears for incident energy above 708 eV, at which the other excitation centred at 90 meV emerges. The 90-meV excitation has a full-width at half-maximum nearly equal to the instrumental energy resolution and resonates at 708.4 eV. The distinct incident X-ray energies for these resonant excitations indicate that the 200- and 90-meV features arise from $Fe^{2+}$ and $Fe^{3+}$ states, respectively.

Many experimental[35–37] and theoretical[30,38–41] studies have shown that $L$-edge RIXS allows spin–flip processes that are not accessible with optical spectroscopy[23]. For example, if both the incident and scattered X-rays are $\pi$-polarized, the spin–flip excitation of $d_{yz}$ is allowed owing to the spin–orbit coupling in the $2p$ core state. In the present RIXS measurements with a 90°-scattering geometry (see Supplementary Fig. 2a), the intensity of elastic excitation with incident X-rays of $\pi$ polarization is reduced in comparison with that of $\sigma$ polarization, and spin–flip excitations are effectively revealed. In addition, the cross-section

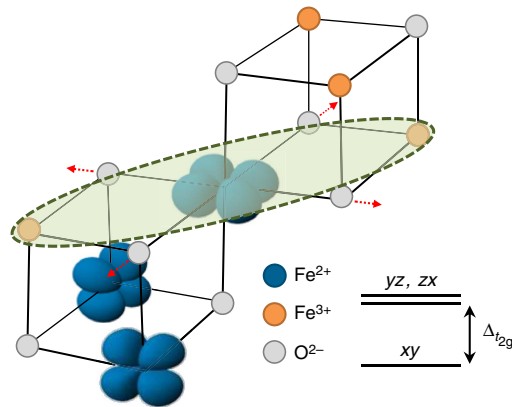

**Figure 1 | Trimeron scenario and $t_{2g}$ energy-level splitting.** Illustration of the orbital ordering of $B$-site $Fe^{2+}$ in $Fe_3O_4$ and the corresponding $t_{2g}$ energy-level splitting for a $Fe^{2+}$ ion in a negative $\Delta_{t_{2g}}$ crystal field. A trimeron is indicated with a dashed oval. The elongation of the four Fe–O bonds in the $xy$ plane are indicated with arrows.

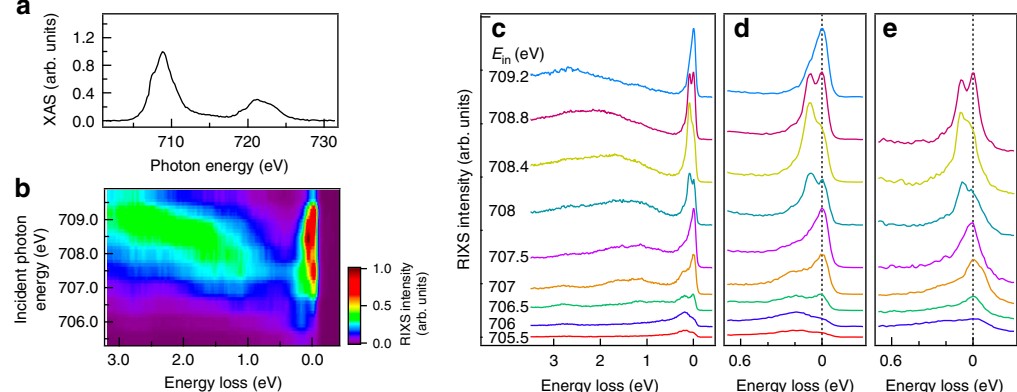

**Figure 2 | RIXS measurements of Fe₃O₄.** (**a**) Fe $L$-edge X-ray absorption spectrum (XAS) spectrum measured in the fluorescence yield mode through the summation of all inelastic X-ray intensities taken at room temperature $T = 300$ K. The XAS is plotted with correction for self-absorption. The incident X-ray energy resolution was 0.5 eV. (**b**) Colour map of RIXS intensity after correction for self-absorption in the plane of incident photon energy versus energy loss recorded at $T = 80$ K. (**c–e**) RIXS spectra plotted in terms of energy loss with a vertical offset for clarity. They were recorded by using $\pi$-polarized incident X-rays under the scattering geometry of the scattering angle $\phi = 90°$ and the incident angle $\phi = 20°$. Panels **c,d** were measured at 80 K and **e** was measured at 550 K.

of $L_3$-edge RIXS for a magnetic excitation is usually larger than that for a phonon excitation[35]. Because O $K$-edge RIXS-probe excitations derived from bimagnons[42] and phonons[43], we performed O $K$-edge RIXS measurements to probe the phonons of Fe₃O₄. Our data shown in Supplementary Fig. 4 reveal an excitation at 70 meV in the O $K$-edge RIXS, indicating that the observed 90-meV feature of the Fe $L_3$-edge RIXS has a small contribution from phonon excitation[44,45].

**Multiplet RIXS calculations.** In order to characterize the origin of the observed excitations, we undertook crystal-field multiplet calculations for the $B$-site $Fe^{3+}$ and $Fe^{2+}$ ionic configurations. See the Methods section, Supplementary Note 2 and Supplementary Figs 5 and 6 for calculation details.

Multiplet calculations carried out for the $B$-site $Fe^{3+}$ ions under an exchange molecular field of 90 meV, as shown in Supplementary Fig. 5d, explain the observed excitation energy of 90 meV well. This is consistent with the 100-meV Zeeman splitting induced by the molecular field deduced from the Curie temperature of magnetite and the exchange coupling constants[46]. This spin–flip energy also agrees with the energy of the nearly dispersionless mode at 80–85 meV observed in inelastic neutron scattering[46,47]. For an individual $Fe^{3+}$ site, the intensity of the 90-meV RIXS feature can change under a spin reorientation as magnetic RIXS is sensitive to the spin direction with respect to the incident polarization. However, RIXS measurements of $B$-sites $Fe^{3+}$ reflect an average of eight non-equivalent $Fe^{3+}$. Each of these non-equivalent $Fe^{3+}$ $B$-sites can make different contributions to the intensity of the 90-meV feature, and our measured RIXS spectra suggest that these changes are beyond our experimental sensitivity. Although we cannot totally rule out phonon contributions, the 90-meV $L_3$-edge RIXS excitation is best explained as a result of spin–flip excitations of $Fe^{3+}$ ions, like magnetic excitations observed in the RIXS of Fe pnictide superconductors[36], cuprates[30,37,40] and nickelates[38,41].

The ground state of the octahedral $Fe^{2+}$ ion is a high-spin $^5T_{2g}$ state with $S = 2$. According to Hund's rule, out of the six $3d$ electrons of the $Fe^{2+}$ ion, five $3d$ electrons occupy spin-up states $t_{2g}^{3\uparrow} e_g^{2\uparrow}$; the remaining one electron occupies one of the three spin-down orbitals $t_{2g}^{1\downarrow}$. When the spin–orbit effect of $3d$ electrons couple a pseudo-orbital angular momentum $\tilde{L} = 1$ to $S = 2$, the $^5T_{2g}$ state splits into three manifolds of pseudo-angular momenta

$\tilde{J} = 1$, 2 and 3. That is, there are effectively 15 separate states from $Fe^{2+}$, as the $^5T_{2g}$ ground state is split by the combination of these interactions. For the broad 200-meV RIXS feature associated with the octahedral $Fe^{2+}$ states, the excitation energy is too large to be explained in terms of spin–flip excitations only. We carefully checked the effect of the local Jahn-Teller distortion to explain the energy of the observed excitations to understand the nature of this feature. From an extensive set of RIXS calculations of $Fe^{2+}$ with varied tetragonal distortions as shown in Supplementary Fig. 6, we found that the average RIXS spectrum calculated using $H_{ex} = 90$ meV and $\Delta_{t_{2g}} = -22$, $-26$ and $-30$ meV explains the measured spectrum most satisfactorily, as demonstrated in Supplementary Fig. 5f. Figure 3a shows the calculated low-energy RIXS excitations of $Fe^{2+}$ in the form of incident photon energy versus energy-loss maps. The calculated RIXS obtained by including the tetragonal distortion, exchange interaction and $3d$ spin–orbit coupling matches fairly well with the experimental data.

**Discussion**

In comparison with the magnified intensity map of RIXS measurements shown in Fig. 3b, calculations using a molecular field $H_{ex} = 90$ meV and $\Delta_{t_{2g}} = -26 \pm 4$ meV reproduce the energy-loss features suitably. For the 200-meV excitation, the experimental resonance starts at an energy lower than that of the resonant quasi-elastic scattering of $Fe^{2+}$ and its energy range is broad, while the calculated resonance starts at a higher energy with a narrow range. This discrepancy is attributed to differences in the dynamics of $3d$ orbitals due to core–hole effects in intermediate states. These effects do not affect the energy loss of excitation spectra because the core holes are filled in the RIXS final state. As is typical of RIXS calculations reported in the literature, our calculations do not include such core–hole effects, and hence do not reproduce the incident energy dependence perfectly, but our calculations correctly reproduce the energy-loss features.

Figure 3c presents calculated RIXS spectrum in comparison with measurements of the incident X-ray energy set to 707 eV, at which the 200-meV RIXS feature is most pronounced. The negative value of $\Delta_{t_{2g}}$ signifies that the energy of $d_{xy}$ is lower than that of $d_{yz}/d_{zx}$, that is, tetragonally distorted $Fe^{2+}O_6$ octahedra with elongated Fe–O bonds in the $xy$ plane. This shows that the tetragonal distortion is directly related to a polaronic distortion of

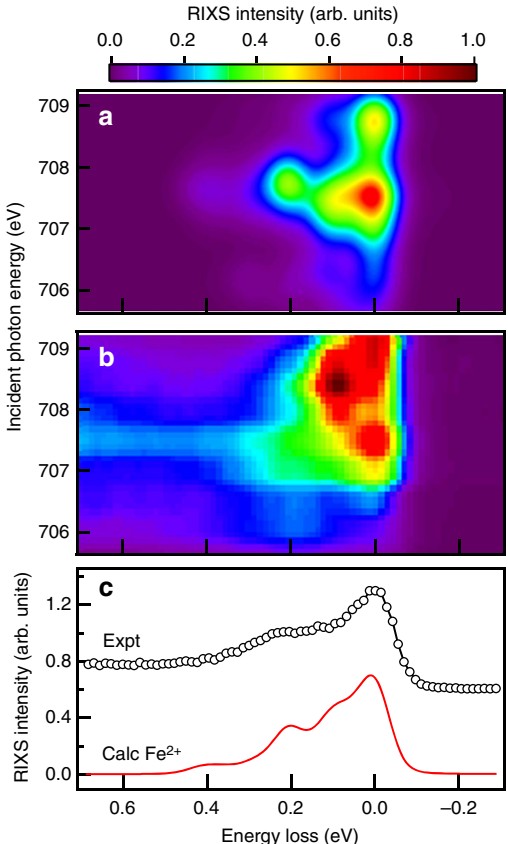

**Figure 3 | Calculated RIXS of Fe$^{2+}$ in comparison with measurements.**
(**a**) Calculated RIXS intensity map of B-site Fe$^{2+}$ by using $H_{ex} = 90$ meV,
$\Delta_{t_{2g}} = -26 \pm 4$ meV and the 3d spin–orbit coupling $\xi_{3d} = 52$ meV. The
resonance photon energy of Fe$^{2+}$ is set to the experimental resonance
energy 707.5 eV. The core–hole lifetime width is set to 200 meV, and the
final-state lifetime width is set to 10 meV. This calculated intensity map
presents the average RIXS intensity for the magnetic easy axis along the
[100], [010] and [001] directions, and are plotted after Gaussian
broadening of width 500 and 80 meV for the incident photon energy and
the energy loss, respectively. (**b**) A magnified intensity map of RIXS
measurements of single-crystal Fe$_3$O$_4$ extracted from Fig. 2b. (**c**) Comparison
of measured (expt) and calculated (calc) RIXS spectra. Open circles are
measurements with incident X-rays of 707 eV at 80 K; the solid line presents
the calculated RIXS spectra of incident X-ray energy 707.5 eV.

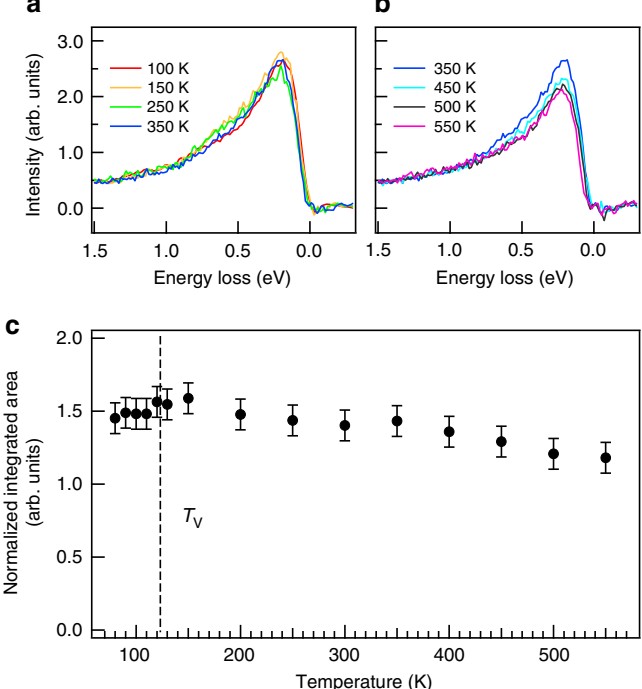

**Figure 4 | Temperature-dependent spin–orbital excitations of Fe$_3$O$_4$.**
(**a,b**) RIXS spectra after the subtraction of the elastic component at
selected temperatures. The spectra were recorded with the incident X-ray
energy set to 706 eV. (**c**) Plot of the integrated 200-meV RIXS intensity
versus temperature. The data were analysed by subtracting the elastic
component from the measured spectrum and normalizing to the intensity of
the dd excitation feature at 2.8 eV (see Supplementary Fig. 9). The dashed
line indicates the Verwey transition temperature $T_V$. The RIXS data
comprise an average of four runs of experimental results. The error bars are
deduced from the average value of variations in the spectral intensity of
four sets of measurements.

the Fe$^{2+}$O$_6$ octahedra, which in turn couple to the neighbouring
Fe$^{3+}$O$_6$ octahedra constituting the trimerons, although, as
mentioned earlier, they are Jahn-Teller-inactive in the first
approximation. Our results are consistent with the locally
distorted structure of the FeO$_6$ octahedra and the short-range
order above $T_V$ observed by X-ray absorption[48] and diffuse
scattering[29]. These short-range correlations are polaronic in
nature[23,26,27,29,49]. Because a local molecular field and a
tetragonal Jahn-Teller polaronic distortion are required to
correctly simulate the excitation energy, the observed spin–
orbital excitations are, indeed, magnetic polarons.

The magnitude of obtained $\Delta_{t_{2g}}$ is comparable with the 3d
spin–orbit coupling strength, and thus confirms the observation
of the unquenched orbital moment[50], which is known from work
on Fe$^{2+}$ impurities in MgO thin films[51]. These results are
also consistent with conclusions of band-structure calculations
using the monoclinic P2/c crystal structure of magnetite[9]
(see Supplementary Note 3 and Supplementary Fig. 7), which
give an energy splitting $\sim 50$ meV between minority-spin $d_{xy}$ and
$d_{yz}/d_{zx}$ bands at the $\Gamma$ point, conforming to the deduced $\Delta_{t_{2g}}$.

We also performed RIXS measurements above the Verwey
transition and found that the spin–orbital excitations driven by
polaronic distortion do exist in the cubic phase of Fe$_3$O$_4$ at high
temperatures as shown in Fig. 2e and Supplementary Fig. 8.
Figure 4 plots the temperature-dependent RIXS spectra with the
incident X-ray energy set to the pre-edge absorption at 706 eV, an
incident X-ray energy at which the elastic component is weak and
the RIXS arises predominantly from octahedral Fe$^{2+}$ ions with a
negligible contribution from Fe$^{3+}$. The temperature-dependent
results show that, when the temperature is varied across $T_V$, the
spin–orbital excitation of 200 meV does not abruptly change its
intensity and persists at least up to 550 K, albeit with a gradual
decrease above 350 K. We interpret this as a gradual weakening of
the polarons. RIXS results shown here serve as a fast probe to
snapshot the dynamic lattice–spin–orbital excitations of Fe$_3$O$_4$.
These temperature-dependent RIXS results indicate that the
FeO$_6$ octahedra are already locally distorted in the cubic phase
of magnetite, in good agreement with the existence of the
short-range correlations in the lattice structure above $T_V$. The
temperature dependence of these distortions follows that of the
magnetization of magnetite, suggesting short-range ordering
of the Jahn-Teller distortion, which gets weakened as the
temperature approaches the Curie temperature $T_C$, and providing
further evidence for magnetic polarons. These observations
suggest that the local distortion in the cubic phase could be
attributed to the precursor of the monoclinic phase across the
Verwey transition.

To summarize, our results demonstrate the usefulness of RIXS to unravel the local electronic structure of a mixed-valence compound by selecting the energy and polarization of incident X-rays. We revealed *dd* excitons in magnetite that have an energy centroid 200 meV and arise from polaronic distortion-driven spin–orbital excitations, which are best explained as magnetic polarons. We also applied crystal-field multiplet calculations to obtain the $t_{2g}$ crystal field $\Delta_{t_{2g}} = -26 \pm 4$ meV induced by the tetragonal Jahn-Teller distortion. These results are consistent with the mechanism of ordering trimerons for the Verwey transition. It would be interesting to carry out RIXS experiments with an improved energy resolution to study the change of spin–orbital excitations across the Verwey transition.

## Methods

**RIXS measurements.** Using the AGM–AGS spectrometer at beamline 05A1 of the Taiwan Light Source[31], we measured RIXS on a single-crystal $Fe_3O_4$(001) at incident photon energies set to specific energies about the $L_3(2p_{3/2} \rightarrow 3d)$ absorption edge of Fe. See Supplementary Fig. 2a for the scattering geometry. Both the scattering angle $\phi$ defined as the angle between the incident and the scattered X-rays, and the incident angle $\theta$ from the crystal *ab* plane, were variable. The polarization of the incident X-ray was switchable between $\pi$ and $\sigma$ polarizations, that is, the polarization within and perpendicular to the scattering plane, respectively, and the polarization of scattered X-rays was not analysed. The energy bandwidth of the incident X-rays was 500 meV, and the total RIXS energy resolution was ~80 meV because the energy compensation method was used to ensure a high-resolution measurement in the energy-loss scheme[31]. The beam diameter of incident X-ray at the sample is ~0.5 mm.

**Sample preparation.** Single-crystal growth of magnetite was carried out in an infrared image furnace in high-purity argon gas (99.999% purity) atmosphere. Measurements of the temperature-dependent specific heat and resistivity of the synthesized magnetite crystal showed that it exhibits a sharp first-order Verwey transition at $T_V = 122$ K. The synthesized single crystal has a chemical composition of $Fe_{3(1-\delta)}O_4$ with $|\delta| \leq 0.00018$, indicative of a nearly ideal chemical stoichiometry. See Supplementary Figs 10 and 11 for the sample characterization.

**Multiplet calculations.** We undertook crystal-field multiplet RIXS calculations of *B*-site $Fe^{2+}$ and $Fe^{3+}$ using CTM4RIXS[52] and MISSING (Dallera and Gusmeroli http://www.esrf.eu/computing/scientic/MISSING/) with the scattering angle 90° and the magnetization axis perpendicular to the scattering plane or in the scattering plane with angles 20° or 70° to the incident beam. The polarization of incident X-ray was selected to be $\pi$-polarized. Since the polarization of scattered X-rays was not analysed in the measurements, we summed calculated RIXS spectra of scattered X-rays with $\sigma$ and $\pi$ polarizations. We used a Lorentzian broadening 0.2 eV for the lifetime width of the intermediate states. The calculated spectra are obtained as an average of the spectra calculated for magnetic domains with the easy axis along the [100], [010] and [001] directions. The crystal-field parameter 10Dq was set to 1.13 eV, and the Slater integrals were reduced to 79% of their atomic values for accurately reproducing the *dd* excitation energies.

**Data availability.** The data that support the findings of this study are available from the corresponding authors on request.

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

## Acknowledgements

We thank A. Fujimori, J. Paul Attfield, Jeroen van de Brink, Sumio Ishihara, Chun-Fu Chang, Maurits Haverkort and Hebatalla Elnaggar for valuable discussions. We acknowledge Martin Beye for sharing unpublished results with us. This work was supported in part by the Ministry of Science and Technology of Taiwan under Grant No. 103-2112-M-213-008-MY3. J.-S.Z. was supported by the DOD-ARMY grant (W911NF-16-1-0559) in USA. J.-G.P. was supported by the research programme of the Institute for Basic Science (IBS-R009-G1) in Korea.

## Author contributions

All authors made significant contributions. H.Y.H., Z.Y.C., W.B.W., J.O. and A.S. performed RIXS measurements. H.Y.H., R.-P.W. and F.M.F.d.G. performed multiplet calculations. H.-T.J. and G.Y.G. performed band-structure calculations. C.T.C. designed the RIXS beamline and spectrometer. Z.-Y.L and J.-S.Z. synthesized and prepared the magnetite single crystals. D.J.H., H.Y.H., F.M.F.d.G., A.C., J.-S.Z., J.-G.P. and L.H.T. analysed the data, discussed the results and wrote the paper. D.J.H. is responsible for project planning.

## Additional information

**Competing interests:** The authors declare no competing financial interests.

**Publisher's note**: 

