## [Peer Review File · Nature Communications]

REVIEWERS' COMMENTS:

Reviewer #3 (Remarks to the Author):

I thank the authors again for providing clarification and answers to the points addressed in my last report. Overall, the provided replies helped me improving the understanding of the study presented by the authors on the magnetite.

I consider the work suitable for publication in Nature Communications, if the comments below can be taken into account. While the authors provided supporting arguments in their replies which answered my questions, I would like to see few more changes reflected in the paper as well, so that the content of the manuscript could be more transparent to the general reader.

1) Can the author mention in the manuscript what explained in their reply "We agree that, for an individual Fe³⁺ site, the intensity of the 90-meV RIXS feature can change under a spin reorientation as magnetic RIXS is sensitive to the spin direction with respect to the incident polarization. However, RIXS measurements of Fe³⁺ reflect an average of 8 non-equivalent Fe³⁺+B-sites, ... These non-equivalent Fe³⁺ B-sites can each make different contributions to the intensity of the 90-meV feature, and our measured RIXS spectra suggest these changes are beyond our experimental sensitivity."

2) Can the author rephrase the sentence in the manuscript "The cross section of L3-edge RIXS for a magnetic excitation is much larger than that for a phonon excitation" by replacing "much" with "usually" or something with similar meaning?

3) Can the author add a reference to the new Sec. D and Fig. S11 of the Suppl. Inf. in the second paragraph of Pag. 7 "The magnitude of obtained Δt_{2g} ...?"

Response to the Referees

(Original reviewers comments are in black italic fonts; our responses are in blue.)

Reviewer #3 (Remarks to the Author):

I thank the authors again for providing clarification and answers to the points addressed in my last report. Overall, the provided replies helped me improving the understanding of the study presented by the authors on the magnetite.

I consider the work suitable for publication in Nature Communications, if the comments below can be taken into account. While the authors provided supporting arguments in their replies which answered my questions, I would like to see few more changes reflected in the paper as well, so that the content of the manuscript could be more transparent to the general reader.

1) Can the author mention in the manuscript what explained in their reply “We agree that, for an individual Fe³⁺ site, the intensity of the 90-meV RIXS feature can change under a spin reorientation as magnetic RIXS is sensitive to the spin direction with respect to the incident polarization. However, RIXS measurements of Fe³⁺ reflect an average of 8 non-equivalent Fe³⁺B-sites, ... These non-equivalent Fe³⁺ B-sites can each make different contributions to the intensity of the 90-meV feature, and our measured RIXS spectra suggest these changes are beyond our experimental sensitivity.”

Our response: We sincerely thank the Referee for valuable comments which helped us to improve the manuscript and for recommending publication of our manuscript. We have included these new three sentences to the paper as suggested by the Referee.

2) Can the author rephrase the sentence in the manuscript “The cross section of L3-edge RIXS for a magnetic excitation is much larger than that for a phonon excitation” by replacing “much” with “usually” or something with similar meaning?

Our response: We have changed “much” to “usually.”

3) Can the author add a reference to the new Sec. D and Fig. S11 of the Suppl. Inf. in the second paragraph of Pag. 7 “The magnitude of obtained Δt_{2g} ...”?

Our response: A reference to Supplementary Note 3 and Supplementary Fig. 7 has been added to the 3rd paragraph of the **Discussion** section.